# Hybrid Cellular Automata Modeling Reveals the Effects of Glucose Gradients on Tumour Spheroid Growth

**DOI:** 10.3390/cancers15235660

**Published:** 2023-11-30

**Authors:** Luca Messina, Rosalia Ferraro, Maria J. Peláez, Zhihui Wang, Vittorio Cristini, Prashant Dogra, Sergio Caserta

**Affiliations:** 1Dipartimento di Ingegneria Chimica, dei Materiali e della Produzione Industriale, Università degli Studi di Napoli Federico II, 80125 Naples, Italy; luca.messina@unina.it (L.M.); rosalia.ferraro@unina.it (R.F.); 2CEINGE-Biotecnologie Avanzate Franco Salvatore, Via G. Salvatore 436, 80131 Naples, Italy; 3Mathematics in Medicine Program, Houston Methodist Research Institute, Houston, TX 77030, USA; mjp11@rice.edu (M.J.P.); zwang@houstonmethodist.org (Z.W.); vcristini@houstonmethodist.org (V.C.); 4Department of Physiology and Biophysics, Weill Cornell Medical College, New York, NY 10065, USA; 5Neal Cancer Center, Houston Methodist Research Institute, Houston, TX 77030, USA; 6Department of Imaging Physics, University of Texas M.D. Anderson Cancer Center, Houston, TX 77030, USA; 7Physiology, Biophysics, and Systems Biology Program, Graduate School of Medical Sciences, Weill Cornell Medicine, New York, NY 10065, USA

**Keywords:** hybrid cellular automata, tumor spheroid, agent-based modeling, cancer

## Abstract

**Simple Summary:**

In recent years, mathematical models have revolutionized cancer research, illuminating the complex dynamics of tumor growth and aiding drug development. These models, reflecting biological and physical processes, are increasingly used in clinical practice, offering precise patient-specific predictions. Our work introduces an innovative in silico model to simulate tumor growth and invasiveness. The automated hybrid cell, replicating key tumor cell features, enables exploration of 3D tumor spheroid evolution. Sensitivity analyses reveal that tumor growth is primarily influenced by cell replication speed and adhesion, while invasiveness relies on chemotaxis. These insights shed light on tumor development mechanisms, guiding effective strategies against tumor progression. Our model serves as a valuable tool for advancing cancer biology research and potential therapeutic interventions.

**Abstract:**

Purpose: In recent years, mathematical models have become instrumental in cancer research, offering insights into tumor growth dynamics, and guiding the development of pharmacological strategies. These models, encompassing diverse biological and physical processes, are increasingly used in clinical settings, showing remarkable predictive precision for individual patient outcomes and therapeutic responses. Methods: Motivated by these advancements, our study introduces an innovative in silico model for simulating tumor growth and invasiveness. The automated hybrid cell emulates critical tumor cell characteristics, including rapid proliferation, heightened motility, reduced cell adhesion, and increased responsiveness to chemotactic signals. This model explores the potential evolution of 3D tumor spheroids by manipulating biological parameters and microenvironment factors, focusing on nutrient availability. Results: Our comprehensive global and local sensitivity analysis reveals that tumor growth primarily depends on cell duplication speed and cell-to-cell adhesion, rather than external chemical gradients. Conversely, tumor invasiveness is predominantly driven by chemotaxis. These insights illuminate tumor development mechanisms, providing vital guidance for effective strategies against tumor progression. Our proposed model is a valuable tool for advancing cancer biology research and exploring potential therapeutic interventions.

## 1. Introduction

According to the World Health Organization, cancer remains one of the leading causes of death in advanced countries. In 2020 alone, there were 19.3 million new cases and 10 million deaths reported worldwide [1,2]. However, there has been a consistent decrease in cancer-related mortality, primarily due to successful prevention efforts and advancements in treatment options [3]. It is noteworthy that approximately 66.7% of cancer deaths are related to a process known as metastasis [4], which refers to the formation of secondary tumors in parts of the body different from the site of cancer origin. The process involves cancer cells crossing endothelial walls, circulating through in the bloodstream, and eventually extravasating from capillaries crossing the basement membrane to colonize the new site [5].

A key feature of the abovementioned steps is the ability of cancer cells to migrate directionally in response to external stimuli, known as chemotaxis, which plays a fundamental role in enabling tumor invasiveness [6]. Chemotaxis is defined as the directional movement of an organism in response to the concentration gradient of a given chemical species. In the context of cell chemotaxis, it refers to the sensitivity of cells to specific gradients, quantified as the ratio between the concentration gradient and the local concentration value (SG=∇C/C) [7,8]. Chemokines (e.g., interleukin) and growth factors (e.g., FBS) are examples of chemical species that drive chemotaxis in cancer cells [7]. Metabolites, such as glucose or oxygen, which are also present in many growth factors, as well as catabolite gradients, can also induce chemotaxis.

In the case of cancer tissues, cell over-proliferation induces a lack of nutrients and an accumulation of catabolites. Consequently, tumor masses lose compactness and, in order to keep a high surface to volume ratio, tend to invade the surrounding tissues. This morphologic instability [9], known as diffusional instability [10,11], can be predicted by mathematical models. The investigation of such a complex phenomenon requires the use of adequate models able to include both biological and transport phenomena aspects. Different approaches have been followed in cancer research, ranging from simple 2D cell cultures, complex 3D scaffolds in vitro, murine in vivo models, to clinical studies.

In cancer research, in vitro approaches, which involve the study of biological systems in artificial laboratory conditions, have been widely employed to investigate the underlying mechanisms of cell migration, proliferation, and other processes related to tumorigenesis. Most of the published data regarding known cell-based processes are derived from experiments performed in two-dimensional (2D) conditions. Two-dimensional cell cultures, growing on solid substrates, such as plastic, do not fully reproduce the complex three-dimensional (3D) architecture and complexity of living tissues. Important aspects, such as cell–cell interactions, tissue phenotypes, the role of cell density and extracellular matrix (ECM) [12,13,14,15], proliferation regulators [16], and metabolic functions [17] are often missing or not adequately represented in 2D cultures.

In order to mimic the native in vivo scenario [18,19,20] where cancer growth occurs, 3D models have been used in cancer research as a compromise between 2D cell cultures and whole-animal systems. However, 3D tumor microenvironments in murine models may not accurately represent the human scenario, making it challenging to control and interpret experimental outcomes.

Recently, the tumor spheroid [8], a tightly bound aggregate of cells, is gaining popularity as a 3D model due to of its ability to strikingly mirror the 3D cellular context. Being characterized by naturally physiological and chemical gradients, cellular spheroids consist of actively proliferating cells on the outside with quiescent cells in the inner rim, and a central nutrient-deprived zone (necrotic core) [21], mimicking the natural scenario in a tumor in situ.

Each of the abovementioned approaches has limitations that can be taken into account by adequately coupling experimental investigations with in silico mathematical models. One of the main challenges is to capture the complex and dynamic interactions between cancer cells and their microenvironment. In vitro experimental conditions may introduce artifacts that can potentially compromise the validity of conclusions drawn from these studies if not properly accounted for. In silico approaches, based on mathematical modeling and computer simulations, have the potential to overcome these limitations by focusing on the representation of the interactions of cells with the microenvironment, which are difficult to mimic experimentally [22,23]. Furthermore, in silico models, once developed and adequately validated, can be used to conduct extensive virtual experimental campaigns at a significantly lower cost compared to in vivo and in vitro research. This approach is crucial for the fundamental understanding of mechanisms of cancer growth and can facilitate the simulation of specific treatments on individual patients, or the study of the effects of rare mutations or genetic variations, implementing precision and personalized medicine [24] protocols.

For this reason, in silico approaches [25] are attracting more attention. The earliest attempts to mathematically model tumor growth and invasiveness date back to 1967, and since then, the number and accuracy of these models have continued to grow [14]. These models can be classified into continuum, discrete, and hybrid models [26], and they are employed to simulate and study a wide variety of phenomena, such as the resistance of tumors to different drug treatments [27,28,29], the interaction between tumor cells and their microenvironment [13,15], and the immune system [25,30].

Hybrid agent-based models, such as hybrid cellular automata (HCA), have proven to be effective in investigating complex tumor systems by combining deterministic reaction–diffusion partial differential equations with the representation of the cells as single and autonomous entities (cellular automata, CA [25]) governed by deterministic or stochastic laws. These models consider the interactions between individual cells [26,27,28,29,31] and can provide greater insights into the mechanisms underlying cancer growth and progression, supporting the design of new therapeutic strategies [30,32].

HCA models are particularly useful in many applications in cancer research, where a multi-scales model is required to depict the dynamics of cancer development over time, including the evolution of cell phenotypes and genotypes [25]. These models account for the interaction of individual cells with parameters characterizing the surrounding environment, such as concentration fields or changes in pH [33].

In this study, a computational model based on HCA is developed to simulate the growth and invasiveness of spheroids under different gradients of nutrients. Specifically, chemotaxis is taken into account by estimating the glucose concentration profiles surrounding a cancer cell aggregate (tumor spheroid). The model aims to investigate the roles of cell migration, duplication, and other parameters, such as cell–cell adhesion and sensitivity to chemotactic gradients, in the phenomenon of cancer invasion. A global sensitivity analysis (GSA) and a local sensitivity analysis (LSA) are performed to evaluate the model’s sensitivity to the variations in the abovementioned cellular parameters.

The goal of this study is to develop a model that can capture the complexity of the cancer invasion mechanism, as envisioned by the diffusional instability model [9], while maintaining computational feasibility. This will provide a more comprehensive understanding of cancer biology and aid in the design of new therapeutic strategies.

## 2. Materials and Methods

In the following section, the structure of the hybrid cellular automata model, referred as 2D-HCA, used for simulating the in vitro growth of avascular tumor spheroids, is presented in detail. The model description is organized into several subsections. Firstly, the model domain (Section 2.1.1) and numerical methods used to calculate continuous functions (in our case, glucose concentration) (Section 2.1.2) are presented. The HCA model is described by presenting the cell phenotypes (Section 2.1.3) and rules governing cellular dynamics (Section 2.1.4). Statistical and sensitivity analyses are described in an independent subsection (Section 2.2).

By structuring the model description in this manner, the paper provides a comprehensive overview of the 2D-HCA model and its application in simulating the in vitro growth of avascular tumor spheroids.

### 2.1. Model Development

The 2D domain used in the model represents a layer of the extracellular matrix (ECM) that contains a single tumor spheroid. This domain is discretized into a squared grid, referred to as a lattice. Each element of the lattice (lattice point, LP) can be occupied by a single cell or remain empty, representing the presence of ECM only. The entire grid evolves through a series of discrete time steps, following a set of pre-defined rules. The evolution of each cell is governed by the chemical stimuli in the LP occupied, and by the state of neighboring LPs. In this model, the chemical stimuli are represented by the concentration profile of a chemoattractant, specifically glucose. The model can be viewed as a superposition of two identical square grids, where one represents the cellular layer (see Section 2.1.3) and the other is used to calculate the glucose concentration field (see Section 2.1.2). The model can be easily generalized to account for more chemical species, or other types of stimuli, such as pressure fields, simply adding further layers.

In Figure 1a, a schematic representation (not in scale) of the cellular layer superimposed onto the glucose layer of our model is presented. In the cellular layer, the LPs are colored either in orange or blue, corresponding to the positions occupied or not by the cells. In each LP, the glucose concentration is calculated and schematically represented in Figure 1a in a color scale. As qualitatively reported in Figure 1a, the glucose concentration at the grid margins is higher compared to the center of the domain. The resulting symmetrical radial concentration gradient is achieved due to an isotropic source of glucose from each of the four edges of the domain, and a consumption in the center where the cell spheroid is located.

#### 2.1.1. Domain Building

In this study, the domain considered for the model was a squared layer with dimensions of 2 mm × 2 mm. This domain was discretized into a grid consisting of 100 × 100 LPs. Each LP represented a cell with an approximate diameter of 20 μm. Two scenarios were examined in this study, defined as *isotropic* and *gradient*. Both scenarios shared the same set of parameters, except for the initial and boundary conditions in the glucose layer.

In the isotropic case, the initial glucose concentration was set to 5.5 mM over the entire domain (initial condition), mimicking a physiological level of glucose in the ECM [34]; the concentration was fixed at the boundary of the domain (5.5 mM). As a result, the tumor spheroid placed in the center of the domain was subjected to an isotropic chemical stimulus, analogous to what was qualitatively reported in Figure 1a. In the gradient case, we defined different boundary conditions at the left and right edges of the domain, imposing fixed glucose concentrations of 8 [34,35] and 3 mM [36,37], respectively. As a consequence, the initial condition was a linear concentration gradient $\left(\frac{\partial C(x,y)}{\partial x}=2.5\;mM/m\right)$, with an average concentration (and at the centre of the domain) still equal to 5.5 mM, as in the isotropic case. The initial concentration was constant along the *y*-axis.

The tumor spheroid was initially represented as an aggregate of a few cells, occupying a sub-domain with a radius of 50 μm (21 cells) in the center of the lattice representing the ECM, which, for the sake of simplicity, was assumed to be composed of collagen. Spheroid growth was simulated for 48 h (simulated time). During the experiment, the spheroid evolved while individual cells proliferated and migrated invading the ECM under the chemotactic stimuli of glucose concentration.

#### 2.1.2. Glucose Layer

In the model, glucose was defined as a function of the spatial variable ***x*** = (x,y) and time *t*. The dynamics of the glucose concentration field *C*(***x***,*t*) in time and space is determined by solving the classical Fickian reaction–diffusion equation:(1)$$\frac{\partial C\left(\underline{x},t\right)}{\partial t}=D∙\nabla^{2}C\left(\underline{x},t\right)-k\left(\underline{x},t\right)\;\mathrm{with}\;C\left(\underline{x},0\right)=C_{0}\;\;\;\;\;\;$$
where *D =*
$7∙10^{-10}m^{2}/s$ [38] is the diffusion coefficient of glucose in the collagen-based ECM and *k*(***x****,t*) is the consumption rate of glucose by the cells. The computational cost to obtain the numerical solution of Equation (1) can thus be high, given the considerable disparity between the time scales of cell division (hours to days) and glucose diffusion (seconds); cellular proliferation was treated as an adiabatic perturbation in the chemical field [29]. Thus, using the adiabatic perturbation approximation, Equation (1) is approximated as a pseudo-stationary problem [39]:(2)$$0=D∙\nabla^{2}C\left(\underline{x},t\right)-k\left(\underline{x},t\right)$$
which is numerically solved by the method of simultaneous over-relaxation with the Chebyshev acceleration [29]. This method requires the discretization of the differential equation in terms of finite differences, where *y* and *x* represent the row and column indices of the elements on the grid, and Δ is the size of a single LP (i.e., each of the two edges along *x* and *y*):(3)$$\frac{{(C}_{x+1,y}-C_{x,y})+(C_{x-1,y}-C_{x,y})+(C_{x,y+1}{-C}_{x,y})+(C_{x,y-1}-C_{x,y})}{{∆}^{2}}-\frac{k_{x,y}}{D}=0$$
where Δ is the size of a single LP (in our case 20 μm). Equation (3) can be rearranged by defining its residual ξ_x,y_ as:(4)$${\xi_{x,y}=C}_{x+1,y}+C_{x-1,y}+C_{x,y+1}+C_{x,y-1}-\left(4+\frac{{∆}^{2}k_{x,y}}{D}\right)C_{x,y}$$
and calculating an approximate solution for the concentration field in the subsequent simulation time step t + dt as:(5)$$C_{x,y}^{t+dt}=C_{x,y}^{t}+\frac{\xi_{x,y}^{t}}{\left(4+\frac{{∆}^{2}k_{x,y}}{D}\right)}$$

Glucose consumption by cells (in LPs where cells are present) is calculated according to a Michaelis–Menten kinetics:(6)$$k_{x,y}=V_{max}∙\frac{C_{x,y}}{K_{m}+C_{x,y}}∙I\left(x,y\right)$$
where $V_{max}=0.05\;mol/{(m}^{3}∙s)$ [40] is the maximum consumption rate, $K_{m}=2\;mol/m^{3}$ [40] is the Michaelis–Menten constant, and *I* is a Boolean indicator that takes on the value of 1 or 0 for lattice points that may or may not contain a cell. Because, in our model, two different cell phenotypes (active and starved, see subsequent section) able to consume glucose can occupy LPs, the indicator *I* in Equation (6) is calculated as the sum of two Dirac delta functions, one for each cell phenotype:$$I\left(x,y\right)=\delta \left(x-x_{act},y-y_{act}\right)+\delta \left(x-x_{starv},y-y_{starv}\right)$$
where (*x*_act_, *y*_act_) and (*x*_star_, *y_starv_*) are the coordinates of the LPs where active and starving cells are located. The use of the Dirac delta function allows for the representation of a point-like object, in this case, the presence of a cell in a specific LP.

#### 2.1.3. Cellular Layer and Cell Phenotypes

In the cellular layer of our model, the dynamics of individual cells were governed by specific rules that governed cellular dynamics and phenotyping, including migration, proliferation, starvation, and death.

Cell time evolution was governed by the nutrient condition. In the cellular layer, three different cellular phenotypes were distinguished: active, starving (or quiescent), and necrotic cells. The occurrence of various phenotypes depended on the cellular microenvironment and affected the metabolic activity of the cells. Two critical glucose concentrations defined the threshold for starving and necrotic cells (C_starv_ > C_nec_). A cell was active if the local glucose concentration *C*_x,y_ (i.e., concentration in the LP occupied) was higher than both these predefined thresholds, *C*_x,y_ > *C*_starv_ > *C*_nec_. An active cell consumed glucose according to Equation (6) and its metabolism included the possibility to migrate and eventually duplicate upon the completion of its cell cycle.

If *C*_nec_ < *C*_x,y_ < *C*_starv_, the cells did not have enough nutrients to duplicate, and thus starved. The cells still consumed residual glucose and could migrate, looking for more nutrient-rich LPs. As time progressed, if *C*_x,y_ increased back to levels higher than *C*_starv_, a starved cell could become active again, unless it remained in the starved state for too long a period, going in apoptosis (see subsequent section).

If the nutrient concentration decreased further, *C*_x,y_ < *C*_nec_, the cell underwent necrosis. This process is irreversible, independent of any future change in the glucose concentration, and from that time on, the necrotic cell cannot proliferate, migrate, or consume glucose.

#### 2.1.4. Rules Governing Cellular Dynamics

The dynamic evolution of cells is highly dependent on nutrient availability, particularly glucose concentration. In response to nutrient availability, cells can experience different fates, including necrosis and apoptosis (cell death).

Each non-necrotic cell in the cellular layer can undergo migration or (if active) proliferation. The two mechanisms are regulated by two independent characteristic times, defined as the migration time *T_m_* and duplication time *T_d_*.

(a)Cell migration

As the simulation time progresses, at every time interval *T_m_*, a cell can migrate. The probability of the cell to migrate is affected by cell–cell interactions if the cell under evaluation is attached to a cell cluster and not isolated (detached). If a migration event occurs, the cell occupies one of its eight neighboring LPs, provided the target location is empty (Figure 1b); alternatively, if migration does not occur, the cell remains in the same position until the subsequent time interval, *T_m_*.

When a cell tries to migrate, it must overcome cell adhesion defined by a probability *P*_adh_, defined as:(7)$$\left\{\begin{array}{l} P_{adh}=\frac{N_{cell}^{att}}{K+N_{cell}^{att}}\;\;\mathrm{if}\;\mathrm{attached}\\ P_{adh}=0\;\;\;\;\mathrm{if}\;\mathrm{attached} \end{array}\right.$$
where $N_{cell}^{att}$ is the number of attached cells confined with the cell of interest ($0<N_{cell}^{att}<8$), while *K* $\left(K\in R\right)$ is a parameter quantifying the role of cell–cell adhesion. For *K* = 0, cell–cell adhesion is supposed to be strong enough to inhibit the possibility of any movement to the cells, while high values of *K* are related to a weak cell–cell interaction and the higher motility of the cells. When an attached cell attempts to migrate, a random number $\left(0<\sigma <1\right)$ is generated and compared to $P_{adh}$. If $\sigma <P_{adh}$, the migration fails, until the subsequent time interval, *T_m_*. If $\sigma >P_{adh}$, migration is allowed and a change in the cell adhesion state can be induced; in our model, for simplicity, we assume that cell detachment is irreversible, i.e., once detached, a cell is not allowed to re-attach. If the cell of interest is already detached $\left(P_{adh}=0\right)$, it shows the typical behavior of isolated cells and is expected to migrate every *T_m_*.

If migration is allowed, the cell has to choose a motility direction, which is defined according to a biased random walk approach. Each of the nine possible positions $P_{i}(i=1,2,\ldots ,9)$ that a cell can take (Figure 1b), including the position already occupied (P_5_), is associated with a numeric interval $R_{i}=[a_{i-1},\;a_{i}[$. The 9 intervals have different sizes $\left(\left|R_{i}\right|=a_{i}-a_{i-1}\right)$, are contiguous and non overlapping, and span the entire range 0–1 ($0<\left|R_{i}\right|<1,\;\sum_{i=1}^{9}\left|R_{i}\right|=1)$ (Figure 1c). A further random number 0≤λ≤1 is generated and compared to the $R_{i}$ intervals; if $\lambda \in R_{i}$, the cell migrates towards direction $P_{i}$, if the corresponding LP is empty. In particular, if $\lambda \in R_{5}$, i.e., the LP already occupied by the cell, the cell does not migrate. If the cell, upon arrival at its new location, is not contiguous with any attached cells, it undergoes a transition into the detached state, provided it is not already in that state.

To evaluate the range of $R_{i}$ and define the probability of migration in each direction, a score $S_{i}$ is evaluated for each of the 9 candidate positions as:(8)$$S_{i}=e^{\alpha A_{i}}$$
where α∈R is a parameter describing the glucose chemotactic sensitivity of the cell, and $A_{i}$ is the local glucose specific gradient, defined as $A_{i}=\left(C_{i}-C_{5}\right)/C_{5}$. $S_{i}$ values are further normalized ${\bar{s}}_{i}=\frac{S_{i}}{\sum_{j=1}^{9}S_{j}}$. Finally, the ranges $R_{i}=\left[a_{i},\;a_{i+1}\right]$ are calculated as $a_{i}=\sum_{j=1}^{i-1}{\bar{s}}_{j}$ (Figure 1c).

(b)Cell proliferation

As the simulation time progresses, at every time interval *T_d_*, active cells proliferate, generating a new daughter cell, identical to its progenitor. The new cell is randomly located in one of the free positions among the eight LPs surrounding the progenitor. If no empty spot is available, the new cell’s location is chosen by identifying the direction where the minimum number of cells separate the progenitor from the edge of the cluster. All the cells along the selected direction shift one position away from the progenitor cell, and the new-born cell occupies the vacancy. It is worth mentioning our model considered the memory of the history of each cell to consider the possible cell-dependent variables, such as the random mutations of cell parameters, even if, in this work, this feature was not used. In each time step, all the cells were considered the same, with the only differences being among the active, starved, and necrotic phenotypes, while cell dynamic evolution was dependent on the nutrient availability only. The code implemented in our model also allowed for more complex interactions. Another simplification of our model was that only active cells attached to the spheroid could duplicate, while detached cells were expected to enter irreversibly in a migration state, unless nutrient availability induced their deaths.

(c)Cell death

In our model, if a cell remained in starved state for a time longer than the apoptosis time (*T*_apop_), it could spontaneously die. This biological event, known as apoptosis [41], is a mechanism of defense of the cells to prevent the propagation of lesions to the future generation. In our model, apoptosis corresponded to the degradation of the cell, which was dissolved in the ECM, and left the previously occupied LP empty. It is worth mentioning that, in the case where the glucose concentration reduces further to values below *C*_nec_, while the cell is already in a starved state, the cell does not enter a state of apoptosis, but evolves into a necrotic state, where it remains indefinitely. Necrotic cells, according to our model, do not proliferate nor migrate, but continue indefinitely to occupy the same LP. The flowchart of the whole cell dynamic algorithm is presented in Figure 2. The model implementation of cell dynamics is also summarized as a sequence of operations in the figure caption.

#### 2.1.5. Analysis

In this study, we focused on analyzing the effects of four key input parameters on the dynamic evolution of cell spheroids in our model: $T_{d}$, $T_{m}$, K, and α. By investigating the influence of these parameters, we aimed to gain insights into the roles of cell adhesion, proliferation, and migration in tumor growth and invasion processes.

In Table 1, the output variables computed in our model are briefly summarized and they include the number of cells attached within the spheroid ($N_{cell}^{core}$), the number of cells that migrated away from the spheroid ($N_{cell}^{mig}$), and the total number of cells ($N_{cell}^{TOT}$). Additionally, the percentages of adhered and migrated cells relative to the total number were calculated as $N_{\%}^{core}$ and $N_{\%}^{mig}$, respectively. The ratio of adhered to migrated cells, ϕ, was also determined by calculating the ratio of $N_{cell}^{core}/N_{cell}^{mig}$, and considered as a measure of the invasiveness of the tumor. The lower the ϕ value, the higher the tendency of the cancer cells to invade and colonize the surrounding ECM.

Furthermore, the spatial domain where the spheroid was located was divided into four sectors according to the diagonals of the square domain and named as North (N), South (S), East (E), and West (W) (Appendix A). The number of migrated cells located in each of these four quadrants were counted and represented as $N_{cell}^{N}$, $N_{cell}^{S}$, $N_{cell}^{E}$, and $N_{cell}^{W}$. The corresponding percentages of migrated cells in each quadrant relative to the total number of migrated cells ($N_{cell}^{mig}$) were also calculated.

### 2.2. Statistical Analysis

In this study, we employed statistical analysis techniques to achieve a greater understanding of our model by conducting both global and local sensitivity analysis. GSA enabled us to assess the collective impact of perturbing all input parameters simultaneously, providing insights into how variations in multiple parameters influenced the system as a whole, while LSA allowed us to investigate the effects of perturbing individual parameters, providing detailed information on the sensitivity of the system to specific parameter changes. This approach allowed us to identify the parameters that had a greater impact on specific aspect of the model, and to identify the controlling mechanism that drove the entire dynamic of the system. In our model, we investigated, in particular, the effect of varying the input parameters related to cell duplication time, cell migration, chemotactic motility, and cell–cell adhesion: *T_d_*, *T_m_*, *K*, and *α*.

#### 2.2.1. Average and Variance Convergence

Since stochastic events and the values of randomly generated numbers governed our HCA model, each simulation run was unique, even if the values of the parameters, initial conditions, and boundary conditions were kept the same. For this reason, the outcomes of one single simulation for a given set of the four input parameters (*T_d_*, *T_m_*, *K*, and *α*) were not enough to guarantee the statistical significance of the results. Therefore, for each set of input parameters, the simulation was replicated n times, defining the actual outcomes as the arithmetic average of the *n* iteration. To identify a value of n reliable from the statistical point of view, a convergence analysis was performed to study the effect of the number of simulation replicates on the model outputs (see Appendix A, Appendix A). Convergence was verified on 4 different sets of model parameters, for the gradient experimental condition, running up to 100 simulations for each set. For *n* ≅ 10, all the model outcomes became constant and independent on *n* (data reported in Appendix A) for each of the 4 sets of parameters investigated. Therefore, in the results presented below, each simulation was reiterated and mediated for *n* = 40.

#### 2.2.2. Global and Sensitivity Analyses

To explore the impact of simultaneous parameter perturbations on the key model outputs, a GSA was conducted, according to the protocols reported in the literature [42,43,44]. In brief, Latin hypercube sampling (LHS) was used to sample the multidimensional parameter space generating a set of 500 combinations of the 4 key parameters of interest (α ∈ [0, 30], *K* ∈ [0, 4], T_d_ ∈ [10, 60 h] [45,46], and *T_m_* ∈ [10, 120 min] [47]).

As there is no literature available to guide the choice of ranges for α and K, the selection was based on informed judgment and the biological context. For the α parameter, which was an index of cell chemotaxis, the lower limit of 0 represented a situation where cells did not sense concentration gradients, while the positive range reflected a scenario where the cells were attracted to increasing glucose concentrations. The upper limit of 30 was chosen to provide a degree of certainty that most cells would localize in regions that were rich in nutrients. For the *K* parameter, which was an index of the cell–cell adhesion strength, the lower limit of 0 corresponded to the strongest cell adhesion that inhibited any cell movement, while the upper limit of 4 was arbitrarily chosen to represent an adhesion strength weak enough for all cells to detach from the spheroid in a time frame of 48 h.

For each of the 500 combinations of the parameters, a simulation of *n* = 10 iterations was run. A multivariate linear regression analysis (MLRA) was performed using the built-in MATLAB function *mvregress* on a whole set of samples to obtain the linear regression coefficients for each parameter. The MLRA technique is based on the idea to express the output of interest $Y_{i}$ as a linear function of the input parameters $X_{j},$ as shown in Equation (9):(9)$$Y_{i}=\varepsilon_{i}+\beta_{1}X_{1}+\beta_{2}X_{2}+\beta_{3}X_{3}+\ldots =\varepsilon_{i}+\sum_{j}\beta_{j}X_{j}$$

$\varepsilon_{i}$ is an error term and $\beta_{j}$ is the regression coefficient of the input parameter. In our case, there were four input parameters $,\;X_{j}$ (T_d_, T_m_, K, and α), and nine outputs, $Y_{i}$, which were $N_{cell}^{core}$, $N_{cell}^{mig}$, $N_{cell}^{E}$, $N_{cell}^{S}$, $N_{cell}^{W}$ $N_{\%}^{core}$, $N_{\%}^{E}$, $N_{\%}^{S}$, and $N_{\%}^{W}$. This subset of outputs, $Y_{i}$, was chosen to be linearly independent, as required by the MLRA. The entire procedure (LHS and MLRA) was repeated 10 times, on 10 different sets of 500 combinations of the input parameter $X_{j}$ value combinations. As result, four regression coefficient distributions were obtained. Each distribution (one for each input parameter, $X_{j}$) represented the measure of the sensitivity index (SI) of the respective parameter.

Finally, a one-way ANOVA test was performed, followed by Tukey’s test (using the built-in MATLAB functions *anova1* and *multcompare*) on the SI distributions. This procedure allowed us to rank the parameters according to their relative significance in affecting the model outputs. The GSA was conducted for both the isotropic and gradient cases.

#### 2.2.3. Local Sensitivity Analysis

Each input parameter was perturbed independently, while keeping the others constant at their respective baseline values. The parameter ranges chosen were the following: chemotactic index α [−30, 30] (basal value α = 6), cell–cell adhesion parameter K [0, 4] (basal value *K* = 0.2), doubling time T_d_ [10, 60 h] (basal value *T_d_* = 18 h), and migration time *T_m_* [10, 120 min] (basal value *T_m_* = 30 min). For α, the range was set between −30 and 30 to account for a chemo-repellent effect. The baseline value of 6 was arbitrarily chosen based on trial and error in order to represent a moderate chemotactic force. For *K*, the range was still between 0 and 4, with a baseline value of 0.2. Furthermore, the baseline value was selected through trial and error to ensure a balanced level of adhesion strength. For each parameter, 150 values were sampled uniformly and randomly within the specified range. Also, in this case, the LSA was conducted for both the isotropic and gradient cases.

## 3. Results

In this section, we present the results of the numerical model simulations, which aim to study the evolution of a tumor spheroid under two different glucose concentration profiles, defined as isotopic and gradient. The results are organized into three main paragraphs. In the first paragraph, we present the results of the simulation of the tumor spheroid’s evolution assuming baseline values for the input parameters of the model. In the second paragraph, we report the result of a GSA used to identify the key parameters that have the most significant impact on the tumor’s evolution. Finally, in the third paragraph, we report the results of an LSA of the parameters to further understand the relationship between the parameters and tumor evolution.

### 3.1. Baseline Case

In this paragraph, we present the results of the simulation of the tumor spheroid’s evolution imposing baseline values onto the four input parameters (i.e., *α* = 6, *K* = 0.2, *T_d_* = 18 h, and *T_m_* = 30 min) both under isotropic and gradient conditions.

Figure 3 displays snapshots of the temporal evolution of the cellular and glucose layers at five time points (0, 12, 24, 36, and 48 h) for both isotropic (Figure 3a,b) and gradient (Figure 3c,d) conditions. The cellular layer (Figure 3a,c) displays the extracellular matrix (ECM) in dark blue, with the spheroid in the center with attached and detached cells, respectively, reported in orange and light blue. The glucose layer (Figure 3b,d) displays the concentration field of glucose in a color scale, ranging from blue (low concentration) to red (high concentration).

The general behavior observed was that the main tumor mass grew over time and consumed nutrients, inducing the formation of concentration gradients from the surrounding environment to the cell-populated area. This chemical stress stimulated cell motion and led to the cells’ detachment and migration toward regions less populated where a higher concentration of nutrients was available, in agreement with the predictions of the diffusional instability theory [9], and its experimental verification [38]. The two conditions investigated presented relevant differences. Under isotropic conditions, the cells moved radially away from the core, without any apparent preferential direction (Figure 3a). Under gradient conditions, as a consequence of the anisotropy in the stimulus, the detached cells tended to travel preferentially toward the source of nutrients at the left edge of the domain (Figure 3c).

It is worth mentioning that, in the base value conditions we investigated, given the limited size of the spheroid and the short time frame simulated, the glucose concentration in the nutrient layer, determined by diffusion and consumption, always remained higher than $C_{starv}=0.1\;mol/m^{3}$. Therefore, all cells remained proliferative, and neither a necrotic core nor a starvation rim were formed in the cellular layer. When running a simulation on a larger domain (5 mm × 5 mm) and for a longer time (20 days), the appearance of necrotic cells was observed, as expected (the results are reported in the Appendix A).

To quantitatively investigate the observed phenomenon and evaluate the numerical measures of tumor evolution over time, we calculated the total number of cells $\left(N_{cell}^{TOT}\right)$ and the number of cells that adhered to the spheroid $\left(N_{cell}^{core}\right)$ as a measure of tumor growth. The ratio of $N_{cell}^{core}$ to cells that migrated away from the spheroid $N_{cell}^{mig}$, defined as ∅, allowed us to compare the proliferation and migration activity, and it was a measure of the invasiveness of the tumour. We independently counted the number of cells that migrated in the four sectors (Appendix A) $\left(N_{cell}^{N},\;N_{cell}^{S},\;N_{cell}^{W},and\;N_{cell}^{E}\right)$ and the respective percentages $\left(N_{\%}^{N},\;N_{\%}^{S},\;N_{\%}^{W},and\;N_{\%}^{E}\right)$. The comparison of these four values was a measurement of anisotropy in cancer invasion and a directional response to chemotactic stimuli (diffusional instability).

Figure 4 presents the time-course plots of isotropic and gradient conditions (Figure 4a–d and e–g, in the top and bottom parts of the figure, respectively). The solid lines report the average output (*n* = 40) values and the ribbons represent the associated standard errors. The orange data report the number (*n* = 40) of cells attached to the spheroid $\left(N_{cell}^{core}\right)$, while the light-blue data represent detached cells $\left(N_{cell}^{mig}\right)$. Figure 4a,e report the total cell count over time, while the graphs in Figure 4b,f report the percentages of the core and migrating cells. Figure 4c,g shows the time-course plot of ratio *ϕ*.

The results show that the tumor core ($N_{cell}^{core}$) has an initial rapid growth rate over time, with an exponential trend for the first 24–36 h; following this initial rapid expansion, $N_{cell}^{core}$ continues to grow almost linearly (Figure 4a,e). The number of cells detaching from the main body and migrating ($N_{cell}^{mig})$ increases linearly over time, starting from the initial value of 0, for the entire temporal window investigated (Figure 4a,e). Accordingly, the percentage of migrated cells ($N_{\%}^{mig})$ increases while the percentage of adhering cells ($N_{\%}^{core})$ decreases as time progresses. After an initial period of instability, both the percentages of adherent and migrated cells reach a steady-state plateau, with the core cells ($N_{\%}^{core})$ representing ~80% of the total number of cells in the domain (Figure 4b,f). This steady state can be easily seen by the constant value of the ratio between the numbers of adherent and migrated cells (ϕ = 4) after 24 h (Figure 4c,g). This result is due to the fact that both the size of the spheroid and the number of migrating cells increase linearly over time, with different rates, for long periods of time. The system, in other terms, reaches a steady state where the generation of cells due to proliferation and the flow of cells leaving the spheroid are balanced.

The abovementioned behavior is comparable in terms of total cells in the core and migrating cells, among the two conditions investigated here, i.e., isotropic and gradient, while a relevant difference can be observed in the direction of migration of the invading cells.

Figure 4d,h report the percentages of migrated cells in the four different directions, defined according to the source of nutrients (see Appendix A). In isotropic conditions, the detached cells do not show any preferential migratory direction and they uniformly distribute into the four sectors of the domain ${(N}_{\%}^{N}≅N_{\%}^{E}≅N_{\%}^{S}≅N_{\%}^{W}=25\%$) (Figure 4d). On the other hand, in the gradient case, a significant difference (*p*<0.005) in the percentage of detached cells that migrate in the four directions is observed. As shown in Figure 4h, the nutrient gradient induces the cells to move in a biased random walkway, such that a higher percentage of the population migrates towards the West (higher glucose concentration). At the end of the simulation (t = 48 h) under anisotropic conditions, 42 ± 12% of cells migrate towards the direction of an increasing concentration gradient, and only 13 ± 8% of the cells move against the gradient, heading towards the East. The remaining migrating cells distribute almost uniformly between the remaining North and South sectors, with 21 ± 8% and 24 ± 10% of cells present in each sector, respectively.

### 3.2. Global and Sensitivity Analysis (GSA)

To perform the analysis, we estimated the SI values of each input parameter ($X_{j}$) and ranked them according to their importance.

In Figure 5, the regression coefficients obtained from the MLRA for the isotropic and gradient cases are plotted. Each bar represents the regression coefficient mean, which is a measure of the SI parameter. The higher the SI, the greater the sensitivity of the parameter in affecting a given model output. To determine the ranking of the relative sensitivity of the model parameters, we conducted a one-way ANOVA test and post hoc analysis (Tukey’s test) after checking that the regression coefficients followed a normal distribution, a crucial requirement for the two tests. The bars marked with an asterisk indicated the statistically significant parameters for each model output, obtained from the MLRA.

From the GSA, we observed that $N_{cell}^{core}$ was independent on α (an analogous result can be obtained by measuring the core area), but it was significantly affected by $T_{d}\;$(Figure 5a,j). The number of migrated cells $N_{cell}^{mig}$ independent on α and $T_{m}$ under both isotropic and gradient conditions (see Figure 5b,k). Looking at the migration direction-related outputs $N_{cell}^{E}$, $N_{cell}^{S}$, $N_{cell}^{W}$, $N_{\%}^{E}$, $N_{\%}^{S}$, and $N_{\%}^{W}$, relevant differences were visible when comparing the two conditions investigated. In the gradient case, all the migration direction-related outputs depended on all four parameters and, in particular, on α (Figure 5l–r). This was expected since α was related to chemotaxis. On the other hand, in the isotropic case, $N_{cell}^{E}$, $N_{cell}^{S}$, and $N_{cell}^{W}$ were only dependent on *T_d_* and *K* (Figure 5c–e), with the former being the most relevant, while $N_{\%}^{E}$, $N_{\%}^{S}$, and $N_{\%}^{W}$ were only affected by $T_{m}$ (Figure 5g–i). From these observations, we can see that α only influences the outputs related to the directionality of cell migration, solely in the presence of a gradient, while it has no effect in isotropic conditions; parameters *K* and *T_d_* affect both the size of the spheroid and the number of cells detaching from it, in both isotropic and gradient conditions; and Tm has a reduced effect on spheroid growth compared to *K* and *T_d_*, but it significantly influences the directionality of cell migration, only in the presence of a gradient.

### 3.3. Local Sensitivity Analysis (LSA)

In order to define the empirical functional relationship between the parameters and model outputs, we conducted an LSA. The results of the LSA provide a greater insight into the non-linear relationships between the parameters and tumor evolution, and highlight the regions of the parameter space where the response variables are most sensitive to changes.

We analyzed the effects of the key input parameters α, K, $T_{d}$, and $T_{m}$ on all model outputs (see Table 1). However, for the sake of brevity, we limited our discussion to only the most significant input–output relationships suggested by the results we obtained from the GSA.

#### 3.3.1. Chemotaxis Sensitivity Index α

As described in the Materials and Methods Section, parameter α was an indicator of cell sensitivity to chemical stimuli, represented by glucose in this work. The higher the absolute value of α, the greater the tendency of the cells to migrate following a given concentration gradient.

Figure 6a,b depict the fraction of migrating cells invading from the spheroid along the four sectors (North, South, West, and East; see Appendix A) of the domain, which are subjected to different chemical stimuli, due to the geometry of our domain. The cells were counted after 48 h of a real-time simulation and reported on the *y*-axis as the fraction of cells in each direction; the *x*-axis showed the α values. The solid lines represent the mean fraction of cells, while the ribbons represent the associated standard deviations calculated for the n = 40 iterations (see Section 2.2.1).

Under isotropic conditions (Figure 6a), the fraction of cells migrating in the four sectors was independent on the α parameter, with the cells uniformly distributed throughout the entire domain ${(N}_{\%}^{N}{≅N}_{\%}^{E}{≅N}_{\%}^{S}≅N_{\%}^{W}≅25\%)$ in agreement with the isotropic stimulus imposed.

In contrast, under anisotropic conditions (Figure 6b), the distribution of migrating cells in the four sectors strongly depended on α. We also investigated the negative values of α to study the potential impact of a chemorepellent (such as a catabolite or toxic drug) on the migration of cancerous cells. Three cases are distinguishable in the graph: α=0, α>0, and α<0.

For α=0, the cells were insensitive to the concentration gradient and the system restored a pseudo-isotropic condition, with migrating cells uniformly distributed ${(N}_{\%}^{N}{≅N}_{\%}^{E}{≅N}_{\%}^{S}≅N_{\%}^{W}≅25\%)$, as in the case of Figure 6a.

For α>0, the cells showed a preferential migration towards the source of the chemoattractant, which was West in our domain. As α increased, $N_{\%}^{W}$ (purple) increased linearly to reach saturation levels for high values of α (α>20), with the fraction of cells plateauing at around 85%. The fraction of cells in the remaining sectors decreased accordingly. It can be seen that the $N_{\%}^{N}$ (light blue) and $N_{\%}^{S}$ (orange) curves almost overlap, due to the North/South symmetry of our setup, while the East direction ($N_{\%}^{E}$, yellow) decreases more rapidly, being opposite to the source of chemoattractant.

A completely inverted situation was observed for α<0, where the cells moved away from the source of the chemical and showed preferential migration towards decreasing gradients, with the directionalities of the cells inverted. The trends of the curves associated with cells migrating towards the west and East were inverted, but the overall trend of the chart was symmetrical.

#### 3.3.2. Cell–Cell Adhesion Parameter *K*

As described in the Materials and Methods Section, parameter K models cell–cell adhesion; the higher the value of K, the greater the tendency of an attached cell to move to one of the adjacent positions. This implies a greater chance that a cell can detach from the spheroid and invade the surrounding environment. Low values of K are related to a limited tendency to invade.

As described in the GSA, parameter K mostly affected the number of cells connected within the spheroid ($N_{cell}^{core}$), the number of detached cells that migrated away ($N_{cell}^{mig}$), and their percentages ($N_{\%}^{core}$ and $N_{\%}^{mig}$). The general result, as expected, was that $N_{\%}^{core}$ monotonically decreased as time progressed and *K* increased.

The trends of $N_{cell}^{core}$ and $N_{cell}^{mig}$ were evaluated at four different time points (12, 24, 36, and 48 h) as a function of parameter K, and are shown in Figure 6c,d, respectively. We reported the data only for the gradient case; the results obtained were independent of the concentration field, as confirmed by the GSA. As in the previous paragraph, solid lines represent the mean values of the cell numbers, while ribbons represent the associated standard deviations.

For simplicity, we distinguished four scenarios: K=0, 0<K<0.5, K=0.5, and K>0.5. When K=0, cell adhesion is indissoluble and, thus, $N_{cell}^{core}$ monotonically increases over time, while $N_{cell}^{mig}$ remains at 0. For values of 0<K<0.5, cell adhesion is still quite strong and $N_{cell}^{core}$ can still increase over time, although at a decreasing rate as K increases. This growth is slowed down by the progressive detachment of migratory cells, which increases over time and as *K* increases.

For the critical value of K=0.5, a steady-state equilibrium was established (Figure 6c, inset). The number of cells adhering to the spheroid remained constant over time and was approximately equal to the initial number of cells (21 cells, t = 0). On the other hand, the number of migratory cells evaluated at 48 h (red) reached its absolute maximum for this value of K. This indicates that there is a balance between the “flow” of cells leaving the tumor and the generation of new cells within the tumor.

For values of K>0.5, cell adhesion weakened further, and the generation of new cells could not compensate for the outflow of migratory cells. The number of adhering cells monotonically decreased over time until the spheroid completely disintegrated at 48 h. Similarly, $N_{cell}^{mig}$ decreased from the peak reached for K=0.5 until it reached a plateau, as there was no longer a tumor core able to generate new cells (Figure 6d). The higher the value of K (above 1), the sooner the tumor core disappears.

#### 3.3.3. Doubling Time *T_d_*

Based on the GSA results, *T_d_* mostly impacts the number of adherents ${(N}_{cell}^{core})$ and migrating cells ${(N}_{cell}^{mig})$. Figure 6e,f display the trends of $N_{cell}^{core}$ and $N_{cell}^{mig}$, evaluated at four different time points (12, 24, 36, and 48 h) as *T_d_* variates. Again, we present the results only for the gradient case. Upon increasing $T_{d}$, the $N_{cell}^{core}$ count (Figure 6e) rapidly decreases, and a similar, but less marked, trend is followed by $N_{cell}^{mig}$ (Figure 6f), reconfirming the results of the GSA.

#### 3.3.4. Migration Time *T_m_*

Once a cell is detached, $T_{m}$ represents the time required by the cell to travel a distance equal to its diameter, thereby indicating the motility of cells. Alternatively, when a cell is attached, its capacity to move is limited, and $T_{m}$ represents the inverse of the frequency at which the attached cell attempts to move.

As described in the GSA, parameter $T_{m}$ mostly affects the fraction of cells comprising the spheroid ($N_{\%}^{core}$) both in isotropic and anisotropic cases, and only in the gradient case is the fraction of migrating cells located in the four sectors ${(N}_{\%}^{N}{,\;N}_{\%}^{E}{,\;N}_{\%}^{S},\;N_{\%}^{W})$.

Figure 6g depicts $N_{\%}^{core}$ evaluated at four time points (12, 24, 36, and 48 h) in gradient conditions. Decreasing $T_{m}$, the core reduces in size because a larger number of cells migrate toward the surroundings (Figure 6g) due to an increased tendency of migration.

Figure 6h shows $N_{\%}^{N}$(light blue), $N_{\%}^{S}$(orange), $N_{\%}^{W}$(purple), and $N_{\%}^{E}$(yellow) evaluated at 48 h as a function of T_m_. In the gradient case, the effect of ${\;T}_{m}$ is observed by an increase in the cell population percentage migrating West (a higher glucose concentration) when ${\;T}_{m}$ decreases (Figure 6h). However, when ${\;T}_{m}$ becomes sufficiently large, the four percentages ($N_{\%}^{N},N_{\%}^{S},N_{\%}^{E},N_{\%}^{W}$) seem to converge to a ~25% value, which can be attributed to the reduced frequency of all cells to migrate.

## 4. Discussion

The HCA model developed in this study successfully simulated the temporal and spatial evolutions of a tumor spheroid under different chemical stimuli. The model incorporated the key biological mechanisms driving tumor progression, such as cell proliferation, migration, apoptosis, and interactions, with the extracellular matrix and neighboring cells. In this study, the growth and invasiveness of an avascular tumor spheroid were simulated in two different chemical fields, isotropic and anisotropic, for 48 h to study its evolution over time. For a fixed set of input parameters, and independent from the type of chemical field, the spheroid core area increased over time, and a fraction of cells detached and invaded the tumor’s surroundings. The effect of an external glucose gradient was to convert cell migration from a random walk to a chemotaxis-driven biased random walk.

Through GSA and LSA, we were able to understand how the selected input parameters affected the system, including how the parameters interacted with each other and how they acted individually. The GSA revealed that the impact of α was negligible in isotropic conditions but was a fundamental factor in governing tumor invasion under the gradient case; parameters Td and K were critical for the evolution of the core area and the number of cells detaching from the spheroid core, in both chemical fields, while *T_m_* had a similar but less effective influence on tumor growth and invasiveness.

The LSA quantitatively assessed the effect of parameter perturbation, showing that the value of α influenced the migration direction of cells towards or away from a positive gradient, while α equal to zero resulted in cells becoming insensitive to the gradient. The parameter K, representing cell–cell adhesion, played a crucial role in tumor evolution, with *K* = 0.5 marking the border between scenarios where adhesion promoted proliferation, or the flux of migrating cells dissolved the tumor over time. Perturbing *T_d_* led to a drastic decrease in the numbers of core and invading cells, while perturbing *T_m_* had the opposite effect, accelerating tumor growth.

Despite its simplicity, the HCA model proved to be highly effective in simulating tumor growth and invasion over time, capturing essential features of tumor biology and reproducing realistic spatiotemporal patterns under different experimental conditions. The model’s versatility allowed for easy modification and adaptation to include additional biological, chemical, and physical processes relevant to tumor growth and invasion. The model could be calibrated and validated using the experimental data obtained from in vitro or in vivo *assays*, thus providing a more accurate representation of the biological system under investigation.

Moreover, the model could be used to design and predict the outcomes of new experiment setups, thereby reducing the need for extensive and costly experiments. By incorporating new biological parameters extracted from simple and rapid experiments, the model could simulate more complex scenarios beyond the limitations of laboratory-based assays. Future improvements can involve the integration of complex signaling pathways, immune responses, hypoxia, nutrient transport, and mechanical stresses, such as oncotic pressure or laminar flows affecting the tumor surface.

In conclusion, the HCA model represents a promising tool for investigating the mechanisms of tumor growth and invasion, as well as for guiding the design and assessment of novel therapeutic strategies. Its ability to capture the complexity of tumor biology and its adaptability to various experimental settings makes it a valuable asset for cancer research.

## 5. Conclusions

This work introduced a model based on the HCA (hybrid cellular automaton) approach, capable of simulating tumor growth and cancer cell invasiveness. The model’s performance was tested, simulating the evolution of a 3D tumoral model, spheroid, at the early stages of its development, and under isotropic and anisotropic glucose concentration fields. Furthermore, this study investigated how perturbations in selected cellular parameters, including chemotactic sensitivity, cellular adhesion, doubling time, and cell motility, affected tumor dynamics.

These parameters were ranked according to their influence on model outcomes by performing a GSA. The results demonstrate that the automata-based model accurately describes the initial exponential-like growth of tumors and depicts cells adapting their migration mechanisms in response to external chemical stimuli. The chemotactic index was identified as crucial for cellular migration in the presence of a chemoattractant gradient, but it was irrelevant for the tumor development in isotropic chemical fields. Cellular adhesion plays an essential role in tumor growth and metastasis. The study revealed an optimal value of parameter *K*, representing cellular adhesion, where the tumor produced the highest number of migrating cells while maintaining a constant size over time. Additionally, the investigation highlighted that a hypothetical cancer cell line characterized by high motility and a short doubling time generated cancers with rapid growth outcomes and high invasiveness.

To ascertain the reliability and utility of our model, its validation through in vitro experiments was essential. For this reason, future developments of this research should primarily focus on validating the model with the empirical data derived from in vitro tumor spheroid growth studies. This crucial step is intended to refine the model’s practicality and expand our comprehension of tumor dynamics. Specifically, in our forthcoming studies, we will analyze the morphological responses of spheroids, focusing on the changes in area and cell detachment when subjected to chemical stimuli. This will include an in-depth investigation of various cell lines, each with unique attributes, such as doubling times, motility values, and cell–cell adhesion forces. These experimental validations will utilize the chemotaxis experiments previously conducted by our research group [38], providing a standardized and replicable framework for our investigations. Supporting this, the preliminary data presented in the Appendix A demonstrate a promising qualitative concordance between the in vitro and in silico outcomes. This alignment pertains to both the morphological response and cell behavior over time, providing a qualitative comparison of the response to the chemoattractant. These experimental validations are anticipated to enhance the model’s accuracy and its capacity to elucidate complex aspects of tumor behavior.

## Figures and Tables

**Figure 1 cancers-15-05660-f001:**
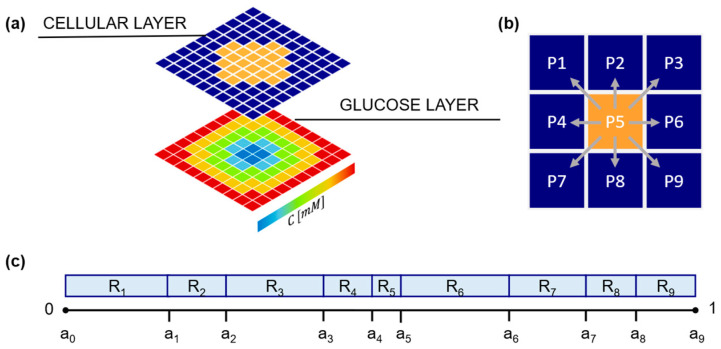
Model schematic showing cellular and glucose field layers. (**a**) Typical initial configuration of the cellular layer (upper grid) with cell-free LPs (dark blue) surrounding cell-occupied LPs (orange). Not in scale. In the glucose layer (bottom grid) glucose concentration is calculated in a pseudo-stationary condition in each lattice point (LP), according to Equation (6). In the scheme, a typical concentration profile in a cell spheroid is reported using a color scale. (**b**) Eight possible migration directions for a representative cell (located in P5); (**c**) schematic representation of the migration direction probabilities (R_i_ values).

**Figure 2 cancers-15-05660-f002:**
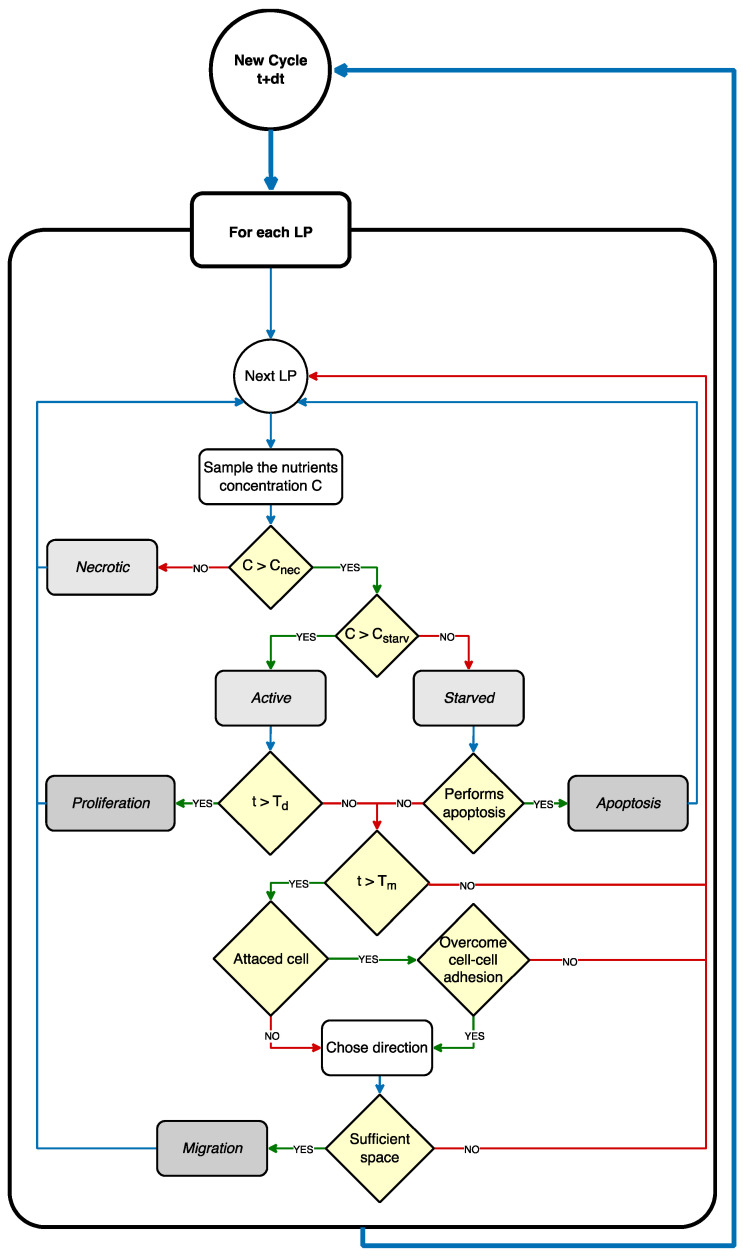
Cell dynamic simulation cycle. The CA model iteratively scans the cellular layer, where each LP represents a cell. If the LP is occupied by a cell, the glucose concentration *C_x,y_* is sampled from the glucose layer; if *C_x,y_ < C_nec_*, the cell is classified as necrotic; if *C_nec_ < C_x,y_ < C_starv_*, the cell is classified as starved; if *C_x,y_ > C_starv_*, the cell is classified as active. Starved cells after *T_apop_* undergo apoptosis and release the occupied LP. Active cells after T_d_ proliferate. Active cells after *T_m_* can migrate, according to the migration rules.

**Figure 3 cancers-15-05660-f003:**
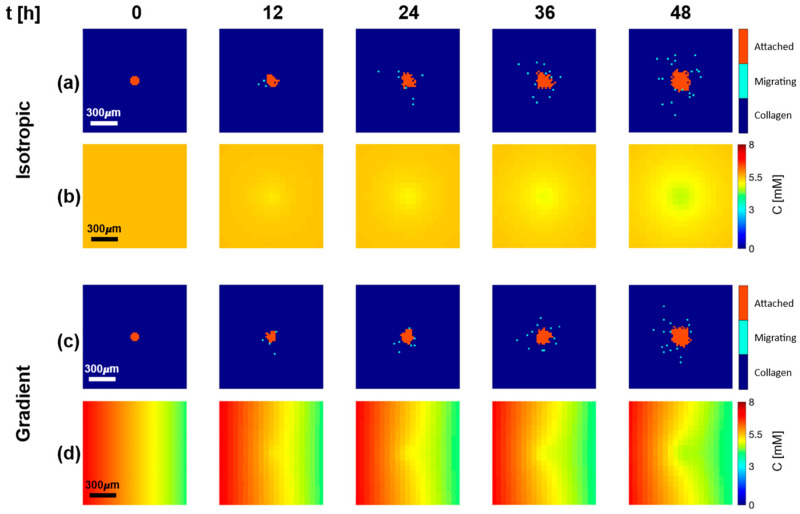
Baseline numerical solution. Snapshots of cellular layer (**a**,**c**) and glucose layer (**b**,**d**) at 0, 12, 24, 36, and 48 h following seeding of cancerous cells in the center of the simulation domain at 0 h, under isotropic (**a**,**b**) and gradient (**c**,**d**) conditions. Cellular layers display cells attached to the core (orange) and cells detached from the core (light blue), which migrate to the collagen matrix (dark blue).

**Figure 4 cancers-15-05660-f004:**
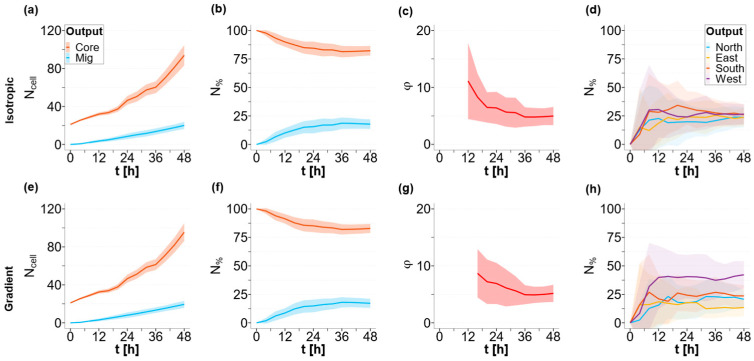
Quantification of baseline solution. Results of the simulation assuming basal conditions with input parameters *α* = 6; *K* = 0.2, *T_d_* = 18 h, *T_m_* = 30 min in isotropic (**a**–**d**) and gradient cases (**e**–**h**); (**a**,**e**) core (orange) and migrated cell numbers over time (light blue); (**b**,**f**) core (orange) and migrated cell percentages (light blue); (**c**,**g**) migrated cell number ratio *ϕ*; (**d**,**h**) percentage of cells migrated in the four directions: North (light blue), East (yellow), South (orange), and West (purple) over time. The solid lines represent the means (n = 40) and the ribbons represents the standard errors.

**Figure 5 cancers-15-05660-f005:**
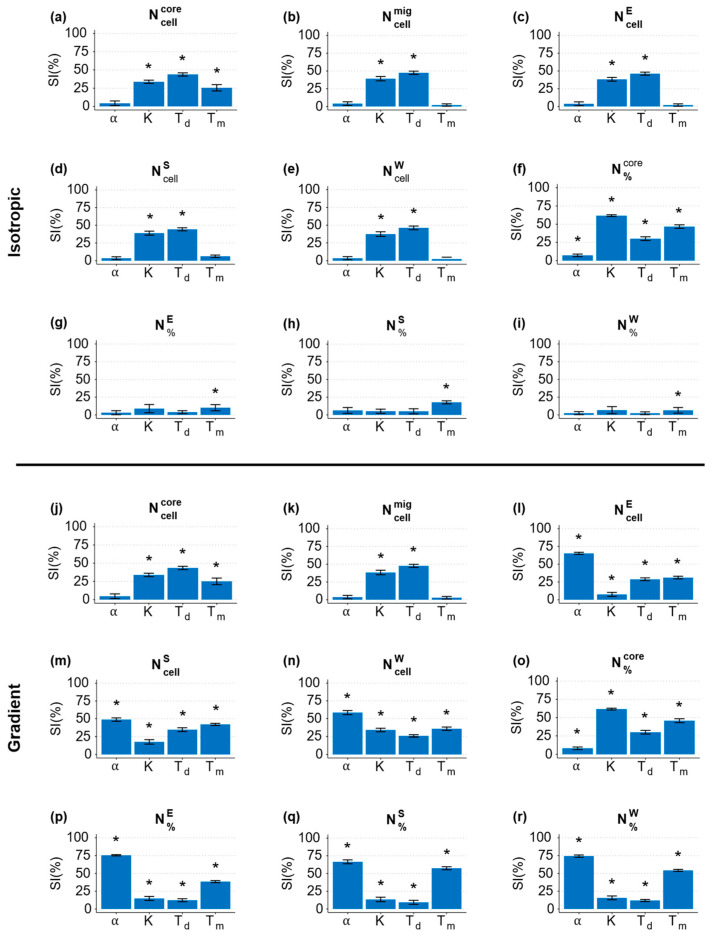
MLRA regression coefficients (SI) in isotropic (**a**–**i**) and gradient (**j**–**r**) cases. The columns refer to the key parameters α, K, $T_{d}$, and $T_{m}$. The rows refer to the outputs of interest: (**a**,**j**) core cell number; (**b**,**k**) migrated cell number; (**c**,**l**) number of cells migrating East; (**d**,**m**) number of cells migrating South; (**e**,**n**) number of cells migrating West; (**f**,**o**) core cell percentage; (**g**,**p**) percentage of cells migrating East; (**h**,**q**) percentage of cells migrating South; (**i,r**) percentage of cells migrating West. The asterisk indicates the significance parameters (*p* < 0.05) for the related outputs.

**Figure 6 cancers-15-05660-f006:**
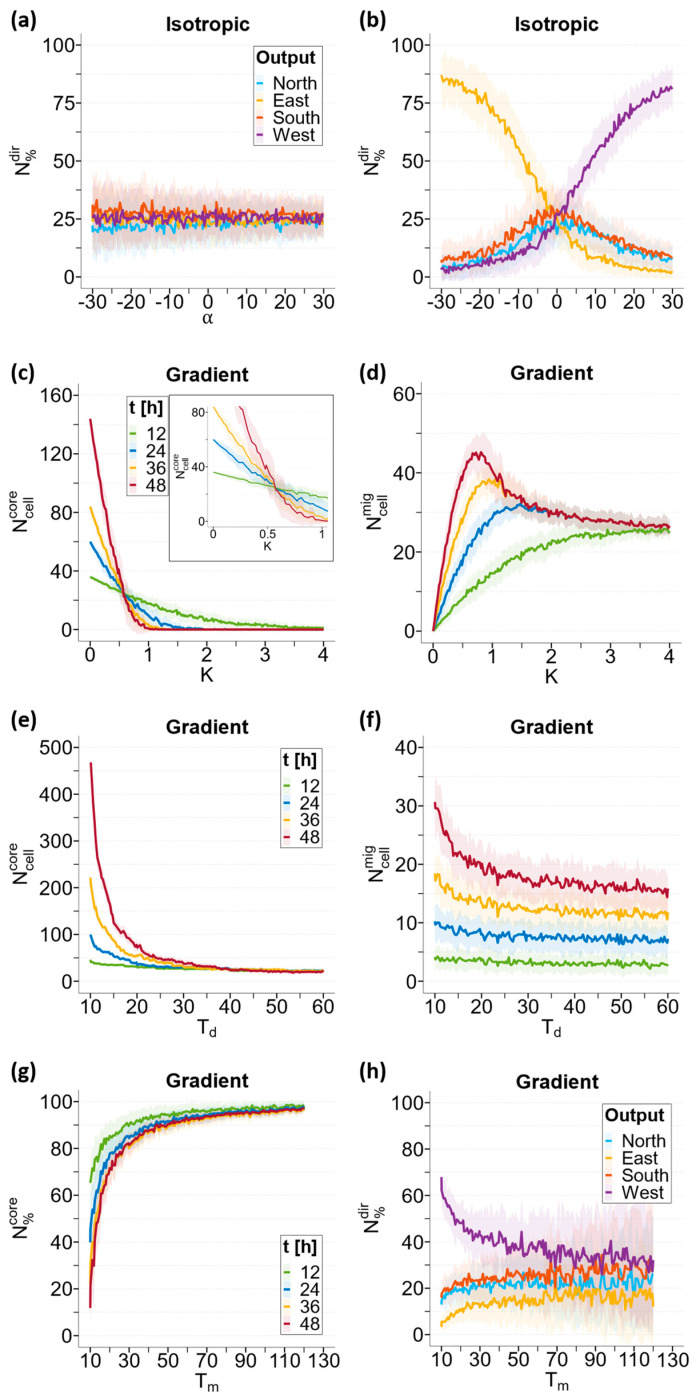
Most relevant LSA results according to the GSA. Results of LSA perturbing input parameter *α*: (**a**) percentages of cells migrating North (light blue), East (yellow), South (orange), and West (purple) at 48 h in the isotropic case; (**b**) percentages of cells migrating North (light blue), East (yellow), South (orange), and West (purple) at 48 h in the gradient case. Results of LSA perturbing input parameter *K*: (**c**) core cell numbers at 12 (green), 24 (blue), 36 (yellow), and 48 h (red) in gradient case; (**d**) migrating cell numbers at 12 (green), 24 (blue), 36 (yellow), and 48 h (red) in gradient case. Results of LSA perturbing input parameter *T*_d_: (**e**) core cell numbers at 12 (green), 24 (blue), 36 (yellow), and 48 h (red); (**f**) migrating cell numbers at 12 (green), 24 (blue), 36 (yellow), and 48 h (red). Results of LSA perturbing input parameter *T*_m_: (**g**) core percentages at 12 (green), 24 (blue), 36 (yellow), and 48 h (red); (**h**) percentages of cells migrating North (light blue), East (yellow), South (orange), and West (purple) at 48 h. The solid lines represent the means (n = 40) and the ribbons represent the standard deviations.

**Table 1 cancers-15-05660-t001:** Model outputs.

Output Variable	Formula	Description
$N_{cell}^{core}$		Number of cells adhered to the spheroid
$N_{cell}^{mig}$		Number of cells that migrated away from the spheroid
$N_{cell}^{TOT}$	$N_{cell}^{core}+N_{cell}^{mig}$	Total number of cells
$N_{\%}^{core}$	$(N_{cell}^{core}/N_{cell}^{TOT})∙100$	Percentage of cells adhered to the spheroid
$N_{\%}^{mig}$	$(N_{cell}^{mig}/N_{cell}^{TOT})∙100$	Percentage of cells that migrated away from the spheroid
ϕ	$N_{cell}^{core}/N_{cell}^{mig}$	Ratio of adhered cells to migrated cells
$N_{cell}^{N}$		Number of migrated cells located in the North quadrant
$N_{cell}^{S}$		Number of migrated cells located in the South quadrant
$N_{cell}^{E}$		Number of migrated cells located in the East quadrant
$N_{cell}^{W}$		Number of migrated cells located in the West quadrant
$N_{\%}^{N}$	$(N_{cell}^{N}/N_{cell}^{mig})∙100$	Percentage of migrated cells located in the North quadrant
$N_{\%}^{S}$	$(N_{cell}^{S}/N_{cell}^{mig})∙100$	Percentage of migrated cells located in the South quadrant
$N_{\%}^{E}$	$(N_{cell}^{E}/N_{cell}^{mig})∙100$	Percentage of migrated cells located in the East quadrant
$N_{\%}^{W}$	$(N_{cell}^{W}/N_{cell}^{mig})∙100$	Percentage of migrated cells located in the West quadrant

## Data Availability

The data presented in this study are available on request from the corresponding author.

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
