# Peer review of "Hybrid Cellular Automata Modeling Reveals the Effects of Glucose Gradients on Tumour Spheroid Growth"

_cancers, 2023, doi:10.3390/cancers15235660_

Round 1

Reviewer 1 Report

Comments and Suggestions for Authors

Overall the authors attempt to model tumor spheroid growth on hybrid glucose gradient. 

The idea of this modeling is unique, yet it is only based on generalized knowledge around cellular biological processes. It would be appropriate to  test this model in a biological setting. 

The statement of "the investigation highlighted that a hypothetical cancer cell line characterized by high motility and a short doubling time generates cancers with fast growth and high invasiveness"  could be easily tested in vitro, in a comparative fashion to a slowly moving cell with high doubling time.

Remeber that many highly migratory cells in culture that have fast doubling times, and are considered metastatic and give rise to these times of cancer.  why does the doubling time matter?

Comments on the Quality of English Language

n/a

Author Response

Overall, the authors attempt to model tumour spheroid growth on hybrid glucose gradient. 

The idea of this modeling is unique, yet it is only based on generalized knowledge around cellular biological processes. It would be appropriate to test this model in a biological setting. 

We appreciate the reviewer's insightful comments. Indeed, our current work primarily focuses on an in silico approach to model tumour spheroid growth on a hybrid glucose gradient. The suggestion to validate this model in a biological setting is well-taken and aligns with our future research direction. Although our manuscript lacks a direct in vitro validation, we have taken steps to address this concern. We have now included a video (S4) as supplementary material, where we provide a qualitative comparison of the morphological response to the chemoattractant in both in silico and in vitro experiments, left and right, respectively. The video demonstrates good agreement between the in vitro and in silico results, both in terms of morphological response and cell behaviour over time.

We believe that this additional material further supports the applicability of our model and its alignment with experimental data. We are committed to continued research in this direction and appreciate the reviewer's feedback.

The statement of "the investigation highlighted that a hypothetical cancer cell line characterized by high motility and a short doubling time generates cancers with fast growth and high invasiveness" could be easily tested in vitro, in a comparative fashion to a slowly moving cell with high doubling time. Remember that many highly migratory cells in culture that have fast doubling times and are considered metastatic and give rise to these times of cancer.  why does the doubling time matter?

We appreciate the reviewer's suggestion and insightful question. The doubling time is indeed a critical parameter to consider in the context of cancer biology. It can significantly impact the overall proliferation rate of a cell population, and this, in turn, may play a pivotal role in cancer progression. We understand the importance of directly testing our hypothesis in vitro by comparing highly migratory cells with a short doubling time to slowly moving cells with a longer doubling time. In our previous work, which was conducted using a 2D model through a wound healing assay (provide citation to preprint), we observed that invasion is more influenced by cell migration than by proliferation. This is an important aspect to consider in our study, and we will explore this further to better understand the interplay between migration, doubling time, and cancer progression in our future research.

Reviewer 2 Report

Comments and Suggestions for Authors
  • The article was nicely designed and presented well. This article clearly explains about how mathematical models have become instrumental in cancer research, offering insights into tumor growth dynamics, and guiding the development of pharmacological strategies.
  • Analysis is clearly explained, detailed description provided in general and gradually going into specific details.
  • Each section is demonstrated with adequate details. Cell dynamic simulation cycle is pictographically well designed and presented.
  • Appropriate references were cited.
  • Abbreviations elaboration in beginning is recommended to remind the reader in the beginning of the article. 
  • The conclusion part can be more specific with additional detailing with respect to the future directives. 
  • doi's are missing for the references
  • article has explained in detail about the HCA model development and the model is unique. 
  • Topic is relevant to the field and would be more compelling if the research is tested in real setting rather than hypothetical settings.
  • Conclusion agrees with method developed and requires some additional information as the proof of concept.
  • References were appropriate with doi's missing.
  • Tables and figures were self-explanatory and were presented well.   

Author Response

The article was nicely designed and presented well. This article clearly explains about how mathematical models have become instrumental in cancer research, offering insights into tumour growth dynamics, and guiding the development of pharmacological strategies. Analysis is clearly explained, detailed description provided in general and gradually going into specific details. Each section is demonstrated with adequate details. Cell dynamic simulation cycle is pictographically well designed and presented. Appropriate references were cited.

We greatly appreciate your positive feedback. We have put significant effort into designing and presenting the article to effectively convey the role of mathematical models in cancer research. Your recognition of the clarity in our explanations and the detail provided is encouraging.

Abbreviations elaboration in beginning is recommended to remind the reader in the beginning of the article.

We appreciate your suggestion. We have made revisions to the article by including expanded explanations for certain abbreviations in the Materials and Methods section, thereby providing a reminder of the definitions established in the Introduction.

The conclusion part can be more specific with additional detailing with respect to the future directives.

We appreciate your feedback. We have made additions to the conclusion section, specifically at line 792, to provide more details regarding future directives and to enhance the specificity of this part.

Doi's are missing for the references article has explained in detail about the HCA model development and the model is unique.

We appreciate your suggestion. We have made revisions to the manuscript, and we have now included the DOI information for the references.

Topic is relevant to the field and would be more compelling if the research is tested in real setting rather than hypothetical settings.

We appreciate the reviewer's insightful comments. Indeed, our current work primarily focuses on an in silico approach to model tumour spheroid growth on a hybrid glucose gradient. The suggestion to validate this model in a biological setting is well-taken and aligns with our future research direction.

Conclusion agrees with method developed and requires some additional information as the proof of concept.

See above.

References were appropriate with DOI's missing.

We appreciate your suggestion. We have made revisions to the manuscript, and we have now included the DOI information for the references.

Tables and figures were self-explanatory and were presented well.  

Thank you!

Round 2

Reviewer 1 Report

Comments and Suggestions for Authors

Overall, the authors attempt to model tumour spheroid growth on hybrid glucose gradient.

The idea of this modeling is unique, yet it is only based on generalized knowledge around cellular biological processes. It would be appropriate to test this model in a biological setting.

We appreciate the reviewer's insightful comments. Indeed, our current work primarily focuses on an in silico approach to model tumour spheroid growth on a hybrid glucose gradient. The suggestion to validate this model in a biological setting is well-taken and aligns with our future research direction. Although our manuscript lacks a direct in vitro validation, we have taken steps to address this concern. We have now included a video (S4) as supplementary material, where we provide a qualitative comparison of the morphological response to the chemoattractant in both in silico and in vitro experiments, left and right, respectively. The video demonstrates good agreement between the in vitro and in silico results, both in terms of morphological response and cell behaviour over time.

----> Again the authors should attempt in vitro verification to support this mathematical model.

We believe that this additional material further supports the applicability of our model and its alignment with experimental data. We are committed to continued research in this direction and appreciate the reviewer's feedback.

The statement of "the investigation highlighted that a hypothetical cancer cell line characterized by high motility and a short doubling time generates cancers with fast growth and high invasiveness" could be easily tested in vitro, in a comparative fashion to a slowly moving cell with high doubling time. Remember that many highly migratory cells in culture that have fast doubling times and are considered metastatic and give rise to these times of cancer. why does the doubling time matter?

We appreciate the reviewer's suggestion and insightful question. The doubling time is indeed a critical parameter to consider in the context of cancer biology. It can significantly impact the overall proliferation rate of a cell population, and this, in turn, may play a pivotal role in cancer progression. We understand the importance of directly testing our hypothesis in vitro by comparing highly migratory cells with a short doubling time to slowly moving cells with a longer doubling time. In our previous work, which was conducted using a 2D model through a wound healing assay (provide citation to preprint), we observed that invasion is more influenced by cell migration than by proliferation. This is an important aspect to consider in our study, and we will explore this further to better understand the interplay between migration, doubling time, and cancer progression in our future research

----> Again the authors should attempt in vitro verification to support this mathematical model.

Comments on the Quality of English Language

----> Again the authors should attempt in vitro verification to support this mathematical model.

Author Response

We have added an explicit statement in the conclusion regarding the need for further validation of the model through in vitro experiments. This addition should provide clarity on the importance of future validation efforts.

Round 3

Reviewer 1 Report

Comments and Suggestions for Authors

Thank you for addressing the authors concerns.

After review, this manuscript belongs in a computational journal   and is not appropriate for cancers. 

Comments on the Quality of English Language

N/A